

# ProCS15: a DFT-based chemical shift predictor for backbone and C$\beta$ atoms in proteins

Anders S. Larsen[1], Lars A. Bratholm[2], Anders S. Christensen[3], Maher Channir[2] and Jan H. Jensen[2]

[1] Department of Pharmacy, University of Copenhagen, Copenhagen, Denmark
[2] Department of Chemistry, University of Copenhagen, Copenhagen, Denmark
[3] Department of Chemistry, University of Wisconsin-Madison, Madison, WI, USA

## ABSTRACT

We present ProCS15: a program that computes the isotropic chemical shielding values of backbone and C$\beta$ atoms given a protein structure in less than a second. ProCS15 is based on around 2.35 million OPBE/6-31G(d,p)//PM6 calculations on tripeptides and small structural models of hydrogen-bonding. The ProCS15-predicted chemical shielding values are compared to experimentally measured chemical shifts for Ubiquitin and the third IgG-binding domain of Protein G through linear regression and yield RMSD values of up to 2.2, 0.7, and 4.8 ppm for carbon, hydrogen, and nitrogen atoms. These RMSD values are very similar to corresponding RMSD values computed using OPBE/6-31G(d,p) for the entire structure for each proteins. These maximum RMSD values can be reduced by using NMR-derived structural ensembles of Ubiquitin. For example, for the largest ensemble the largest RMSD values are 1.7, 0.5, and 3.5 ppm for carbon, hydrogen, and nitrogen. The corresponding RMSD values predicted by several empirical chemical shift predictors range between 0.7–1.1, 0.2–0.4, and 1.8–2.8 ppm for carbon, hydrogen, and nitrogen atoms, respectively.

## INTRODUCTION

Chemical shifts hold valuable structural information that is being used more and more in the determination and refinement of protein structures and dynamics (*Mulder & Filatov, 2010*; *Raman et al., 2010*; *Lange et al., 2012*; *Bratholm et al., 2015*; *Robustelli et al., 2010*) with the aid of empirical shift predictors such as CamShift (*Kohlhoff et al., 2009*), Sparta+ (*Shen & Bax, 2010*), ShiftX2 (*Han et al., 2011*), PPM_One (*Li & Brüschweiler, 2015*) and shAIC (*Nielsen, Eghbalnia & Nielsen, 2012*). These methods are typically based on approximate physical models with adjustable parameters that are optimized by minimizing the discrepancy between experimental and predicted chemical shifts computed using protein structures derived from X-ray crystallography. The agreement with experiment is quite remarkable with RMSD values around 1, 0.3, and 2 ppm for carbon, hydrogen, and nitrogen atoms. Chemical shift predictions based on quantum mechanical (QM)

Corresponding author
Jan H. Jensen,
jhjensen@chem.ku.dk,

calculations (mostly density functional theory, DFT) are becoming increasingly feasible for small proteins (*He, Wang & Merz, 2009*; *Zhu, He & Zhang, 2012*; *Zhu, Zhang & He, 2013*; *Exner et al., 2012*; *Sumowski et al., 2014*; *Swails et al., 2015*) and Vila, Scheraga and co-workers have gone on to develop a DFT-based chemical shift predictor for C$\alpha$ and C$\beta$ atoms called CheShift-2 (*Martin et al., 2013*). Generally, these QM-based methods yield chemical shifts that deviate significantly more from experiment than the empirical methods, with RMSD values that generally are at least twice as large. However, many of these studies have also shown that the empirical methods are less sensitive to the details of the protein geometry and that QM-based chemical shift predictors may be more suitable for protein refinement (*Parker, Houk & Jensen, 2006*; *Sumowski et al., 2014*; *Vila et al., 2009*; *Christensen et al., 2013*).

Some of us recently showed (*Christensen et al., 2013*) that protein refinement using a DFT-based backbone amide proton chemical shift predictor (ProCS) yielded more accurate hydrogen-bond geometries and $^{3h}J_{NC'}$ coupling constants involving backbone amide groups than corresponding refinement with CamShift. Furthermore, the ProCS predictions based on the structurally refined ensemble yielded amide proton chemical shift predictions that were at least as accurate as CamShift. This suggests that the larger RMSD observed for QM-based chemical shift predictions may, at least in part, be due to relatively small errors in the protein structures used for the predictions, and not a deficiency in the choice of DFT functional and basis set. However, in order to test whether this is true in general we need to include the effect of more than one type of chemical shift in the structural refinement. In this study we extend ProCS to the prediction of chemical shifts of backbone and C$\beta$ atoms in a new method we call ProCS15. We describe the underlying theory, which is significantly different from the previous, amide proton-only, version of ProCS (hence the new name) and test the accuracy relative to full DFT calculations as well as experiment for Ubiquitin and the third IgG-binding domain of Protein G (GB3). We also compare the accuracy to CheShift-2 and other commonly used empirical chemical shift predictors using both single structures and NMR-derived ensembles for Ubiquitin.

## THEORY

ProCS15 computes the chemical shift of an atom in residue *i* by

$$\delta^i = b - a\sigma^i \tag{1}$$

where *a* and *b* are empirically determined parameters as discussed further below and $\sigma^i$ is the isotropic chemical shielding of an atom in residue *i*. $\sigma^i$ is computed from the protein structure using the following equation (some of these terms only contribute for certain atom types as described below)

$$\sigma^i = \sigma^i_{BB} + \Delta\sigma^{i-1}_{BB} + \Delta\sigma^{i+1}_{BB} + \Delta\sigma^i_{HB} + \Delta\sigma^i_{H\alpha B} + \Delta\sigma^i_{RC} + \Delta\sigma^i_w. \tag{2}$$

Here $\sigma^i_{BB} = \sigma^i_{BB}(\phi^i, \psi^i, \chi^i_1, \chi^i_2, \ldots)$ is the chemical shielding computed for an Ac-AXA-NMe tripeptide (AXA for short, Fig. 1), where X is residue *i*, for a given combination of $\phi$, $\psi$, and $\chi_1, \chi_2, \ldots$, values as described further in the "Backbone scans" subsection. $\Delta\sigma^{i-1}_{BB}$ is the

**Figure 1 Example of the Ac–AXA–NMe tripeptides (for the case where X = Ser) used to compute the backbone contributions to the chemical shielding values.**

change in chemical shielding of an atom in residue $i$ due to the presence of the side-chain of residue $i-1$. It is computed as

$$\Delta\sigma_{BB}^{i-1} = \sigma_{BB}^{i-1}(\phi^{i-1}, \psi^{i-1}, \chi_1^{i-1}, \chi_2^{i-1}, \ldots) - \sigma^A(\phi_{std}, \psi_{std}). \tag{3}$$

Here $\sigma_{BB}^{i-1}$ is the chemical shielding computed for an AXA tripeptide where X is residue $i-1$, and $\sigma^A$ is from the corresponding calculation on the AAA tripeptide but using $\phi_{std} = -120°$ and $\psi_{std} = 140°$ for all $\phi$ and $\psi$ angles. For example, if residue $i$ is a Ser and residue $i-1$ is a Val then the effect of the Val side-chain on the C$\beta$ chemical shielding of the Ser residue is computed as the difference in the chemical shielding of the C$\beta$ atom in the C-terminal Ala residue computed for an AVA and AAA tripeptide. This approach assumes that the effect of the $i-1$ side chain on the chemical shielding values of the atoms in residue $i$ are independent of the conformations $\phi_i$ and $\psi_i$ angles and the nature of residue $i$. $\sigma_{BB}^{i+1}$ is the corresponding change in chemical shielding of an atom in residue $i$ due to the presence of the side-chain of residue $i+1$.

$\Delta\sigma_{HB}^i$ in Eq. (2) is the effect of hydrogen bonding to the amide H ($\Delta\sigma_{1°HB}^i$) and O ($\Delta\sigma_{2°HB}^i$) atoms of residue $i$ on the chemical shielding of the backbone atoms (this term is zero for C$\beta$)

$$\Delta\sigma_{HB}^i = \Delta\sigma_{1°HB}^i(r_{HO}, \theta, \rho) + \Delta\sigma_{2°HB}^i(r_{OH}, \theta_O, \rho_O). \tag{4}$$

$\Delta\sigma_{1°HB}^i$ is computed using the structural models shown in Fig. 2 as the change in chemical shielding of the backbone atoms in N-methyl acetamide relative to that of the free monomer computed at the OPBE/6–31G(d,p)//PM6 level of theory for a variety of orientations (see the subsection "Hydrogen bond scans" for more information) while the internal monomer geometries are kept fixed. For H$\alpha$ the chemical shielding is taken as the average of the three hydrogen atom on the N-methyl group. Note that the carbonyl carbon formally belongs to residue $i-1$. $\Delta\sigma_{2°HB}^i$ is included only when another amide or amine group is hydrogen bonded to the amide oxygen and is computed as the change in the chemical shielding of the top amide group in Fig. 2A. For H$\alpha$ the chemical shielding is taken as the average of the three hydrogen atoms on the methyl group of the acetamide. Note that in this case the amide nitrogen and hydrogen formally belong to residue $i+1$ and that $r_{HO}$, $\theta$, and $\rho$ are defined relative to the carbonyl oxygen of residue $i$ rather than the amide proton as for $\Delta\sigma_{1°HB}^i$. $r_{HO}$, $\theta$, and $\rho$ are therefore labeled $r_{OH}$, $\theta_O$, and $\rho_O$ in Eq. (4).

**Figure 2 Schematic representation of the model systems used to compute $\Delta\sigma_{HB}^i$.**

**Figure 3 Schematic representation of the model systems used to compute $\Delta\sigma_{H\alpha B}^i$.**

$\Delta\sigma_{H\alpha B}^i$ is the effect of hydrogen bonding to the H$\alpha$ and amide O atoms of residue $i$ on the chemical shielding of the backbone atoms and C$\beta$ and has two contributions

$$\Delta\sigma_{H\alpha B}^i = \Delta\sigma_{1\circ H\alpha B}^i(r_{H\alpha O},\theta,\rho) + \Delta\sigma_{2\circ H\alpha B}^i(r_{OH\alpha},\theta_O,\rho_O). \tag{5}$$

$\Delta\sigma_{1\circ H\alpha B}^i$ is computed using the structural models shown in Fig. 3 as the change in chemical shielding of the backbone and C$\beta$ atoms in Ac-A-NMe relative to that of the free monomer computed at the OPBE/6-31G(d,p)//PM6 level of theory for a variety of orientations (see the subsection "Hydrogen bond scans" for more information) while the internal monomer geometries are kept fixed. $\Delta\sigma_{2\circ HB}^i$ is computed as the change in the chemical shielding of the top amide group in Fig. 3A. For H$\alpha$ the chemical shielding is taken as the average of the three hydrogen atom on the methyl group of the acetamide. Note in this case that the amide nitrogen and hydrogen formally belong to residue $i+1$ and that $r_{HO}$, $\theta$, and $\rho$ are defined relative to the carbonyl oxygen of residue $i$ rather than the amide proton as for $\Delta\sigma_{1\circ HB}^i$. $r_{H\alpha O}$, $\theta$, and $\rho$ are therefore labeled $r_{OH\alpha}$, $\theta_O$, and $\rho_O$ in Eq. (5).

$\Delta\sigma_{RC}^i$ is the effect of ring current on the chemical shielding. Usually this is only significant for proton shift and is thus only calculated for the H$\alpha$ and amide protons.

The ring current is calculated by a simple point-dipole model equation

$$\Delta\sigma_{RC}^i = iB\frac{1 - 3\cos^2(\theta)}{r^3}.$$

The model depends on the parameters $i$, which is the side-chain-specific ring-current intensity relative to benzene, $B$, which is a constant in the model, and the vector $\mathbf{r}$, which is the vector from the proton to the center of the aromatic ring. $\theta$ is the angle between $\mathbf{r}$ and the vector normal to the aromatic ring system. The cut-off for calculating ring current is 8 Å in Procs15 and the value for $i$ and $B$ are taken from *Christensen, Sauer & Jensen (2011)*.

$\Delta\sigma_w^i$ is the change in chemical shielding of an amide proton due to a hydrogen bond to a water molecule. While the backbone terms of ProCS15 is parameterized based on DFT calculations with the polarizable continuum model of solvation, this model does not account for explicit solvent effects and this term is included for amide protons that do not form hydrogen bonds to other atoms in the protein structure. $\Delta\sigma_w^i$ is 2.07 ppm based on DFT calculations on an N-methylacetamide-water complex (*Christensen et al., 2013*).

## METHODOLOGY

### Backbone scans

The capped AXA tripeptides used to compute the first three terms of Eq. (2) were constructed using the FragBuilder Python module (*Christensen, Hamelryck & Jensen, 2014*), which was also used to make different conformations. The acidic and basic amino acids are all modeled in their charged state, including Histidine. This will be the correct charged state for most ionizable residues in most proteins. However, for any ionizable residues that are in their neutral state this approximation can introduce large errors. For example, the $C\beta$ chemical shifts of Asp and His change by 3.0 and 2.4 ppm due to protonation state changes in small peptides, while the N-chemical shifts change by 1.5 and 1.8 ppm (*Platzer, Okon & McIntosh, 2014*). This issue will be addressed in future studies. Only Cysteine is modeled and not the disulfide bonded Cysteine. For each tripeptide a scan on the central residue's backbone and side chain dihedral angles $\phi$, $\psi$, $\chi_1$, $\chi_2$, $\chi_3$, $\chi_4$ was carried out. The $\omega$ dihedral angle was fixed at 180°. The $\phi/\psi$ backbone angles on the N and C-termini alanine residues were fixed at −140° and 120° corresponding to typical $\beta$-sheet residue backbone angles. The scans were done with a 20° grid spacing. For the alanine AAA tripeptide this resulted in 361 conformations from a $\phi/\psi$ scan. For amino acid types with more than two side chain angles this approach would result in far to many samples. Instead we used BASILISK (*Harder et al., 2010*) that allows us to sample from the continuous space of the side chain torsion degrees of freedom. 1,000 conformations were generated for each $\phi/\psi$ backbone pair spaced by 20°. See Table S1 in Supplementary Materials for an overview of the number of conformations sampled for each residue. The geometry of each conformation were optimized with PM6 (*Stewart, 2007*) with the backbone and side chain torsion angles frozen. The GIAO NMR calculations were done at the OPBE/6-31G(d,p) level of theory (*Zhang et al., 2006*) using the CPCM continuum solvation model (*Barone & Coss, 1998*) with a dielectric constant of 78. The rationale for

using 78 is that the bulk solvent effects will have the largest effect for charged side-chains, which are usually located on the surface of the protein. Both the optimization and NMR calculation were done with Gaussian 09 program (*Frisch et al., 2014*). In total the ProCS15 backbone terms are based on ∼2.35 million DFT calculations.

Several structures failed in the optimization stage or had to be discarded due to steric clashes in the NMR calculation and the missing chemical shielding values were found by interpolation. For amino acids with no and one side chain angles cubic interpolation was used and for 2–4 side chain angles nearest neighbor interpolation. For amino acids with 0 side chain angles, the data is interpolated to a grid with 1° grid spacing, 1 side chain angles to a grid of 5° and the rest of the amino acids 20°. The interpolation is done with the Python package SciPy (*Jones, Oliphant & Peterson, 2001*). The grids are saved in the .npy compressed file format from the Numpy Python package. In the compressed state on the hard disk the data size is ∼17 GB and when loaded in to random access memory (RAM) ∼25 GB.

## Hydrogen bond scans

$\Delta\sigma_{HB}$ and $\Delta\sigma_{H\alpha B}$ (cf. Eq. (2)) are parameterized using the model systems shown in Figs. 2 and 3. For $\Delta\sigma_{HB}$ the scans were done by scanning over the hydrogen bond length $r_{OH}$, the bond angle $\theta_H$ defined by H..O= C or H..O–C and the dihedral angle $\rho_H$ defined by H..O= C–N, H..O= C–C or H..O–C(..)H$^O$. The bond length was scanned from 1.5 to 3.0 Å in 0.125 Å steps. $\theta_H$ was scanned from 180.0 to 90.0° in 10.0° steps and $\rho_H$ was done in the entire range −180° to 180°. Similarly, for $\Delta\sigma_{H\alpha B}$ the $r_{OH\alpha}$ bond length was scanned from 1.8 to 4.0 Å in steps of 0.2 Å. The bond angle $\theta_{H\alpha}$ defined by H$\alpha$..O= C or H$\alpha$..O–C was scanned from 180° to 90° with a 10° step size. The $\rho_{H\alpha}$ dihedral H$\alpha$..O= C–N, H$\alpha$..O= C–C and H$\alpha$..O–C(..)H$^O$ was scanned in steps of 15° over the entire range. To get the change in chemical shift caused by the hydrogen bonding the OPBE/6-31G(d,p)//PM6 chemical shielding of systems without hydrogen bonding are subtracted from the scans. The result of the scan is interpolated and saved in another set of .npy files. The monomer geometries are optimized at the PM6 level of theory and kept fixed during the scan.

## NMR calculations and protein structures used

In this paper we benchmark the NMR chemical shift predictions on Ubiquitin and GB3. The structures are geometry optimized using PM6-D3H+ (*Kromann et al., 2014*) using the PCM solvation model (*Tomasi, Mennucci & Cammi, 2005*; *Steinmann et al., 2013*) and the CHARMM22/CMAP force field (*Mackerell, 2004*) using the GB/SA solvation model (*Qiu et al., 1997*) with the 1UBQ (*Vijay-Kumar, Bugg & Cook, 1987*) and 2OED (*Ulmer et al., 2003*) structures as starting points. The PM6-D3H+ optimizations are done using the GAMESS program (*Schmidt et al., 1993*) with a convergence criterion of $5 \times 10^{-4}$ atomic units, while the CHARMM22/CMAP optimizations are done using TINKER (*Ponder & Richards, 1987*) with the default convergence criterion of 0.01 kcal/mole/Å. In addition the following NMR-derived structural ensembles are used without further refinement: 1D3Z (*Cornilescu et al., 1998*), 2K39 (*Lange et al., 2008*), 1XQQ (*Lindorff-Larsen et al., 2005*), 2LJ5 (*Montalvao, Simone & Vendruscolo, 2012*), 2KOX (*Fenwick et al., 2011*). In all

calculations we used charged protonation states for the acidic and basic side-chains, but in the NMR ensembles Histidine was left neutral (with either $N\delta 1$ or $N\varepsilon 2$ protonated) as published. The charges are consistent with the published p$K_a$ values of Ubiquitin (*Sundd et al., 2002*; *Lenkinski et al., 1977*) and GB3 (*Khare et al., 1997*).

OBPE/6-31G(d,p)//PM6-D3H+ GIAO NMR shielding calculations were performed with Gaussian09 using the CPCM solvation model. ProCS15 calculations were done using a module written for the protein simulation framework PHAISTOS (*Boomsma et al., 2013*). The module was specifically written for this paper and can be downloaded at github.com/jensengroup/procs15. CheShift-2 calculations were performed using either the web interface at cheshift.com or the CheShift-2 PyMOL-plugin (*Schrödinger, 2010*) found at github.com/aloctavodia/cheshif. CamShift, PPM_One, Sparta+, shAIC, and ShiftX2 calculations are performed using the stand-alone predictors. The NMR chemical shielding and shifts are compared to shifts measured for Ubiquitin (*Cornilescu et al., 1998*) (BMRB ID 17769)(*Ulrich et al., 2007*) and GB3 (*Vögeli et al., 2012*) (BMRB ID 18531), respectively, both at pH 6.5.

Much of the variation in some of the chemical shifts comes from the nature of the side-chain itself and the side chains before and after in the sequence, which can lead to inflated $r$-values. To separate the contributions of the sequence and the structure we subtract the measured sequence corrected random coil values (*Tamiola, Acar & Mulder, 2010*) from all predicted and experimental values. Note that this does not affect the computed RMSD values.

## RESULTS AND DISCUSSION

### Choice of functional and basis set

When it comes to prediction of chemical shifts in proteins the most widely used functional appears to be B3LYP (*Becke, 1993*). For example, *Zhu, He & Zhang (2012)* used B3LYP/6-31G(d,p) to compute hydrogen and carbon chemical shifts for small proteins that correlate well with experimental measurements with $r$ values typically $\geq 0.98$ when solvent effects are taken into account. Exner, Möller, and co-workers (*2012*) obtained similar results using B3LYP/6-31G(d) and even observed a correlation of 0.81 for the notoriously difficult amide N by averaging over several snapshots. Finally, *Vila, Baldoni & Scheraga (2009)* did a systematic study of the effect of 10 functionals on C$\alpha$ chemical shifts in Ubiquitin and found very little difference in performance with all $r$ and RMSD values in the range 0.902–0.908 and 2.12–2.30 ppm. Interestingly, this study included functionals such as OPBE that are computationally less demanding than B3LYP. Vila, Scheraga and co-workers (*2009*) subsequently observed that C$\alpha$ chemical shifts computed using smaller basis sets such as 6-31G correlate extremely well the chemical shifts computed using lager basis set such as 6-311+G(2d,p). We therefore decided to use the 6-31G(d,p) basis for our calculations and use the computationally efficient OPBE functional.

**Table 1 Comparison of ProCS15 to OPBE/6-31G(d,p)//PM6-D3H+ values computed for the entire protein.** All chemical shielding values are corrected for random coil effects. The RMSD values are computed after linear regression. "All" means that all terms in Eq. (2) are included, with the exception of $\Delta\sigma_w$. "$\Delta\sigma_{BB}^{i-1}$" means that the $\Delta\sigma_{BB}^{i-1}$ term has been removed in the chemical shift prediction, while all other terms are included. The row marked "ProCS15" corresponds to the combination of terms outlined in Table 2.

| | C$\alpha$ | C$\beta$ | C$'$ | H$\alpha$ | H$^N$ | N |
|---|---|---|---|---|---|---|
| | RMSD ($r$) | RMSD ($r$) | RMSD ($r$) | RMSD ($r$) | RMSD ($r$) | RMSD ($r$) |
| **Ubiquitin** | | | | | | |
| All | 1.9 (0.70) | 3.0 (0.50) | 2.1 (0.72) | 0.6 (0.82) | 0.7 (0.85) | 4.9 (0.67) |
| $\Delta\sigma_{BB}^{i-1}$ | 1.9 (0.69) | 3.1 (0.48) | 2.1 (0.72) | 0.6 (0.81) | 0.6 (0.88) | 6.5 (0.50) |
| $\Delta\sigma_{BB}^{i+1}$ | 1.9 (0.71) | 3.1 (0.48) | 2.1 (0.73) | 0.6 (0.82) | 0.7 (0.85) | 5.0 (0.66) |
| $\Delta\sigma_{1\circ HB}^{i}$ | 1.9 (0.72) | – | 2.1 (0.72) | 0.6 (0.82) | 1.3 (0.20) | 4.7 (0.70) |
| $\Delta\sigma_{2\circ HB}^{i}$ | 1.9 (0.69) | – | 2.7 (0.53) | 0.6 (0.80) | 0.8 (0.83) | 5.9 (0.50) |
| $\Delta\sigma_{1\circ H\alpha B}^{i}$ | 1.7 (0.75) | 2.5 (0.69) | 2.1 (0.72) | 1.0 (0.42) | 0.7 (0.86) | 4.4 (0.74) |
| $\Delta\sigma_{2\circ H\alpha B}^{i}$ | 1.9 (0.69) | – | 2.2 (0.71) | 0.6 (0.82) | 0.7 (0.85) | 5.0 (0.66) |
| $\Delta\sigma_{RC}^{i}$ | – | – | – | 0.6 (0.81) | 0.7 (0.85) | – |
| ProCS15 | 1.7 (0.77) | 2.5 (0.69) | 2.1 (0.72) | 0.6 (0.82) | 0.7 (0.85) | 4.4 (0.74) |
| **GB3** | | | | | | |
| All | 1.8 (0.81) | 2.5 (0.58) | 2.4 (0.60) | 0.7 (0.82) | 0.8 (0.82) | 4.7 (0.77) |
| $\Delta\sigma_{BB}^{i-1}$ | 1.7 (0.82) | 2.4 (0.59) | 2.5 (0.52) | 0.7 (0.83) | 0.9 (0.79) | 5.9 (0.61) |
| $\Delta\sigma_{BB}^{i+1}$ | 1.8 (0.81) | 2.4 (0.59) | 2.5 (0.55) | 0.6 (0.84) | 0.8 (0.82) | 4.7 (0.77) |
| $\Delta\sigma_{1\circ HB}^{i}$ | 1.7 (0.84) | – | 2.3 (0.63) | 0.7 (0.83) | 1.4 (0.82) | 5.6 (0.69) |
| $\Delta\sigma_{2\circ HB}^{i}$ | 1.8 (0.80) | – | 2.8 (0.49) | 0.7 (0.81) | 0.8 (0.82) | 5.6 (0.67) |
| $\Delta\sigma_{1\circ H\alpha B}^{i}$ | 1.7 (0.82) | 2.3 (0.60) | 2.4 (0.62) | 1.1 (0.36) | 0.8 (0.82) | 4.5 (0.78) |
| $\Delta\sigma_{2\circ H\alpha B}^{i}$ | 1.8 (0.81) | – | 2.4 (0.60) | 0.7 (0.83) | 0.8 (0.82) | 4.6 (0.77) |
| $\Delta\sigma_{RC}^{i}$ | – | – | – | 0.7 (0.79) | 0.8 (0.80) | – |
| ProCS15 | 1.6 (0.84) | 2.3 (0.60) | 2.3 (0.65) | 0.7 (0.82) | 0.8 (0.82) | 4.5 (0.78) |

## Benchmarking ProCS15 against full QM calculations

Equation (2) is parameterized using OPBE/6-31G(d,p)//PM6 calculations so we compare ProCS15 against full OPBE/6-31G(d,p)//PM6-D3H+ calculations on Ubiquitin (1UBQ) and GB3 (2OED) to test for errors introduced by the inherent additivity assumptions and the structural simplifications in the model systems used for the DFT calculations. We use PM6-D3H+ for the geometry optimization, rather than PM6, to get a better description of hydrogen-bonding and other intermolecular interactions. However, bond lengths and angles, and their effect on chemical shifts, will be very virtually identical to PM6. The results are summarized in Table 1. The first row, marked "all", summarizes results for ProCS15 if all but the last term of Eq. (2) are included. The last term corrects for the explicit solvent effects and thus not relevant when comparing to DFT calculations.

In the case of C$\alpha$ none of the terms have a large effect on the chemical shielding. In the case of GB3 the results improve slightly if $\Delta\sigma_{1\circ HB}^{i}$ is removed and removing $\Delta\sigma_{1\circ H\alpha B}^{i}$ improves the results slightly for both proteins. Accordingly these two terms are removed from ProCS15, while all other terms are kept (note the ring current is only included for hydrogen
**Table 2  Terms in Eq. (2) that are included in ProCS15 for a given atom type are marked with an "x".**

| | $C\alpha$ | $C\beta$ | $C'$ | $H\alpha$ | $H^N$ | $N$ |
|---|---|---|---|---|---|---|
| $\Delta\sigma_{BB}^{i-1}$ | x | x | x | x | x | x |
| $\Delta\sigma_{BB}^{i+1}$ | x | x | x | x | x | x |
| $\Delta\sigma_{1\circ HB}^{i}$ | | | | x | x | x |
| $\Delta\sigma_{2\circ HB}^{i}$ | x | | x | x | x | x |
| $\Delta\sigma_{1\circ H\alpha B}^{i}$ | | | | x | x | |
| $\Delta\sigma_{2\circ H\alpha B}^{i}$ | x | | x | x | x | x |
| $\Delta\sigma_{RC}$ | | | | x | x | |
| $\Delta\sigma_{w}$ | | | | | x | |

atoms). For $C\beta$ removing $\Delta\sigma_{1\circ H\alpha B}^{i}$ decreases the RMSD by 0.2–0.5 ppm, while $\Delta\sigma_{BB}^{i-1}$ and $\Delta\sigma_{BB}^{i+1}$ increases and decreases the RMSD value depending on the protein. Accordingly only $\Delta\sigma_{1\circ H\alpha B}^{i}$ is removed. Note that the structural models used for $\Delta\sigma_{1\circ HB}^{i}$, $\Delta\sigma_{2\circ HB}^{i}$ and $\Delta\sigma_{2\circ H\alpha B}^{i}$ do not contain a $C\beta$ atom so there is no such contribution for this nucleus. For $C'$ removing $\Delta\sigma_{1\circ HB}^{i}$ decreases the RMSD for GB3 by 0.1 ppm so we choose to remove this term for this atom type. Note that removing $\Delta\sigma_{2\circ HB}^{i}$ increases the RMSD by 0.4–0.6 ppm so this term is important for accurate predictions of $C'$ chemical shifts. For $H^N$ and $H\alpha$ we choose to retain all the terms. Not surprisingly, the respective primary hydrogen bonding terms lower the RMSD by 0.4–0.6 ppm and are crucial for accurate predictions. Finally, for N removing $\Delta\sigma_{1\circ H\alpha B}^{i}$ lowers the RMSD by 0.2–0.5 ppm, so this term is removed. Note that $\Delta\sigma_{BB}^{i-1}$ and the two hydrogen bonding terms involving H lower the RMSD by as much as 1.6 ppm ($\Delta\sigma_{BB}^{i-1}$ for Ubiquitin) and is crucial for accurate predictions.

An overview of the terms of Eq. (2) used in ProCS15 for each atom type can be found Table 2 and the RMSD and $r$ values obtained using this combination of terms are given in the row labeled "ProCS15" in Table 1. The RMSD value for the carbon atoms range from 1.6 to 2.5 ppm and a very similar for both proteins. The $r$ values range between 0.60 and 0.84 with the $r$ value being consistently highest for $C\alpha$. For the hydrogen atoms the RMSD and $r$ values range from 0.6 to 0.8 ppm and 0.82 to 0.85, respectively. Finally, for N the RMSD values are 4.3–4.5 ppm, while the $r$ values are in the range 0.74–0.78.

In the case of GB3 the RMSD ($r$) value for $C\beta$ can be reduced (increased) to 1.8 ppm (0.71) by removing a single outlier identified by the Generalized Extreme Studentized Deviate Test (*Rosner, 1983*). The outlier is Ala20 for which ProCS15 and DFT predict a $C\beta$ chemical shielding value of 176.8 and 167.4 ppm, respectively. Inspection of the structure shows that the $C\beta$ atom is only 3.1 Å from the N atom of Ala26—an interaction not taken into account in the parameterization of ProCS15.

Similarly (also for GB3), the RMSD ($r$) value for $H^N$ can be reduced (increased) to 0.6 ppm (0.91) by removing a single outlier identified by the Generalized Extreme Studentized Deviate Test. The outlier is Gln2 for which ProCS15 and DFT predict a $H^N$ chemical shielding value of 24.2 and 20.1 ppm, respectively. Inspection of the structure shows that the $H^N$ atom is within 1.77 Å of the OE1 atom of the Gln2 side chain and within

**Table 3 Comparison of chemical shifts predicted using various methods to experimental values measured for Ubiquitin and GB3 and corrected for random coil effects.** The RMSD values are computed after linear regression. The predictions were done using CHARMM22/CMAP optimized structures using the GB/SA solvation model except for the first two rows (marked with [a]) where PM6-D3H+ optimized structures using the CPCM solvation model were used.

| | $C\alpha$ | $C\beta$ | $C'$ | $H\alpha$ | $H^N$ | $N$ |
|---|---|---|---|---|---|---|
| | RMSD ($r$) | RMSD ($r$) | RMSD ($r$) | RMSD ($r$) | RMSD ($r$) | RMSD ($r$) |
| **Ubiquitin** | | | | | | |
| DFT[a] | 2.1 (0.62) | 2.8 (0.56) | 1.8 (0.85) | 0.4 (0.83) | 0.6 (0.81) | 4.0 (0.80) |
| ProCS15[a] | 2.0 (0.61) | 2.2 (0.52) | 1.7 (0.88) | 0.4 (0.86) | 0.6 (0.73) | 4.4 (0.85) |
| ProCS15 | 1.7 (0.70) | 2.0 (0.50) | 1.7 (0.81) | 0.4 (0.77) | 0.6 (0.72) | 4.0 (0.79) |
| CheShift-2 | 1.7 (0.59) | 1.6 (0.62) | | | | |
| CamShift | 1.1 (0.85) | 1.3 (0.71) | 1.0 (0.81) | 0.3 (0.73) | 0.5 (0.69) | 3.0 (0.63) |
| PPM_One | 0.7 (0.93) | 1.1 (0.80) | 0.9 (0.87) | 0.2 (0.88) | 0.4 (0.73) | 2.2 (0.81) |
| Sparta+ | 0.7 (0.93) | 1.1 (0.82) | 0.8 (0.88) | 0.2 (0.86) | 0.4 (0.72) | 2.2 (0.81) |
| shAIC | 0.7 (0.94) | 1.1 (0.82) | 0.8 (0.89) | 0.3 (0.83) | 0.5 (0.71) | 2.3 (0.79) |
| ShiftX2 | 0.5 (0.97) | 0.7 (0.91) | 0.5 (0.96) | 0.1 (0.97) | 0.3 (0.91) | 1.8 (0.88) |
| **GB3** | | | | | | |
| DFT[a] | 2.1 (0.71) | 2.4 (0.53) | | 0.4 (0.76) | 0.6 (0.86) | 4.6 (0.78) |
| ProCS15[a] | 1.8 (0.73) | 2.1 (0.42) | | 0.4 (0.75) | 0.7 (0.85) | 4.8 (0.88) |
| ProCS15 | 1.6 (0.70) | 2.0 (0.42) | | 0.3 (0.85) | 0.6 (0.76) | 4.3 (0.86) |
| CheShift-2 | 1.7 (0.68) | 1.8 (0.53) | | | | |
| Camshift | 1.2 (0.81) | 1.0 (0.83) | | 0.3 (0.85) | 0.4 (0.82) | 3.3 (0.54) |
| PPM_One | 1.0 (0.87) | 0.9 (0.87) | | 0.3 (0.91) | 0.4 (0.89) | 2.3 (0.79) |
| Sparta+ | 1.0 (0.87) | 1.0 (0.86) | | 0.3 (0.89) | 0.4 (0.88) | 2.8 (0.70) |
| shAIC | 1.0 (0.88) | 1.0 (0.85) | | 0.3 (0.87) | 0.4 (0.83) | 2.3 (0.79) |
| ShiftX2 | 0.6 (0.96) | 0.7 (0.93) | | 0.1 (0.97) | 0.1 (0.98) | 2.3 (0.79) |

2.54 Å of an $H\varepsilon$ atom of the Met1 side chain. While these interactions should be included in the $\sigma_{BB}^i$ and $\Delta\sigma_{BB}^{i-1}$ term, respectively, it is possible that the latter interaction is not found in the scan due to the choice of $\phi_{std}$ and $\psi_{std}$ described above. This residue is also identified as an outlier for N and removing it reduces (increases) the RMSD ($r$) value to 4.1 ppm (0.81).

## Comparison to experimental chemical shifts using single structures

Table 3 shows the comparison of QM, ProCS15 and several common chemical shift predictors to experimental values. The first two rows use the OPBE/6-31G(d,p) and ProCS15 chemical shielding predictions used to construct Table 1 and therefore use the PM6-D3H+optimized structures of Ubiquitin and GB3. However, most future use of ProCS15 will be based on structures optimized with force fields so prediction of the remaining rows is done using structures optimized with the CHARMM22/CMAP force field. The ProCS15 predictions based on the CHARMM22/CMAP-optimized structures include the $\Delta\sigma_w$ term (cf. Eq. (2)). The $a$ and $b$ factors in Eq. (1) are found by linear regression to the experimental values for each atom type. In order to offer a fair comparison RMSD values are computed after a linear fit to the experiment for *all* methods.

The OPBE/6-31G(d,p)//PM6-D3H+ calculations reproduce the experimental chemical shifts to within 2.8 ppm for carbon atoms, 0.6 ppm for hydrogen atoms and 4.6 ppm for nitrogen. The results are similar to those observed by other researchers using other functionals. For example, Zhu and co-workers (*2012*) used B3LYP3/6-31G(d,p)//AMBER (and a locally dense 6-31++G(d,p)/4-31G(d) basis set for C′) and an implicit solvent model to reproduce chemical shift values to within 3.3 ppm for carbon atoms, 0.4 for hydrogen atoms and 8.4 ppm for nitrogen. In this study the RMSD for hydrogen atoms was computed for H$\alpha$ and H$^N$ combined. In a later study (*Zhu, Zhang & He, 2013*), the same researchers reproduced the chemical shifts of amide protons in GB3 to within 0.5 ppm using a locally dense 6-31++G(d,p)/4-31G(d) basis set and a variety of functionals including OPBE. Similarly, Exner and co-workers (*2012*) used B3LYP/6-31G(d)//AMBER and an implicit solvent model to reproduce the H$^N$ chemical shifts of the HA2 Domain to within 0.5 ppm using a single structure and 0.3 pm using several MD snapshots.

While ProCS15 does not reproduce the DFT results perfectly as discussed above the first two rows of Table 3 show that ProCS15 can reproduce experimental chemical shifts with an overall accuracy that is similar to full DFT chemical shielding calculations for Ubiquitin and GB3. The RMSD values predicted with ProCS15 for carbon atoms are 0.1–0.6 ppm lower compared to the DFT results, while the RMSD values for hydrogen and nitrogen atoms are 0.0–0.1 ppm and 0.2–0.4 ppm higher. It is therefore not clear that much is necessarily gained by adding additional terms to ProCS15 without also increasing the underlying level of theory used to compute these terms. For example, it is known that using a larger basis set can significantly improve the prediction of C′ chemical shifts (*Vila et al., 2014*; *Zhu, He & Zhang, 2012*).

Using structures optimized with CHARMM22/CMAP instead of PM6-D3H+ to predict chemical shifts with ProCS15 does also not seem to lead to overall worse agreement with experiment. In fact the results tend to improve slightly (up to 0.5 ppm) for heavy atoms as judged by the RMSD values. Comparison of ProCS15 to CheShift-2, which has also been parameterized against DFT calculations, show fairly similar accuracy for C$\alpha$ and slightly worse accuracy for C$\beta$. The latter observation is perhaps due to the fact that CheShift-2 uses a different (empirical-corrected) reference for each residue type. However, this is also the case for C$\alpha$ for which ProCS15 predictions give a lower RMSD value.

Comparison of ProCS15 to the empirical methods (CamShift through ShiftX2) generally show considerably lower RMSD of the empirical predictions for all atoms types, except H$\alpha$ for GB3 where the accuracy is mostly comparable. The $r$ values are also considerably higher for the empirical methods than for ProCS15 for C$\alpha$ and, especially, C$\beta$, while they are comparable for the remaining atoms.

As mentioned in the introduction the higher RMSD values generally observed for the DFT-based methods compared to the empirical methods is expected. The important issue in the context of structural refinement against measured chemical shifts is whether the DFT-based methods are more sensitive to relative small differences in structure. While a thorough investigation of this complex issue for ProCS15 will be the subject of future studies, we look at the effect of using different structural ensembles on the accuracy next.

## Comparison to experimental chemical shifts using NMR-derived ensembles

Table 4 lists the RMSD and *r* values computed for Ubiquitin using the X-ray structure 1UBQ and five NMR-derived structural ensembles with between 10 and 640 structures. For ProCS15 the average chemical shift is obtained by computing the average chemical shielding for each nucleus followed by the linear regression fit to experimental chemical shift values (cf. Eq. (1)) to obtain the predicted average chemical shifts. The procedure is the same for the remaining methods except that chemical shifts are used instead of chemical shieldings.

For ProCS15 use of ensemble structures lowers the RMSD values for all atom types, with decreases in the range 0.1–0.7 ppm for heavy atoms and 0.1 ppm hydrogen atoms. Similar improvements are observed for C$\alpha$ and C$\beta$ for CheShift-2, except that the improvement in RMSD for C$\beta$ (0.5 ppm) is larger compared to ProCS15 (0.3 ppm). These improvements are expected if the NMR-derived ensembles are a more accurate representation of the protein structure in solution than the single X-ray structure (*Arnautova et al., 2009*; *Vila et al., 2010*). Indeed, all but one of the ensembles used here were generated specifically to be a more realistic presentation of protein ensemble in solutions. The exception is 1D3Z, which is a traditional NMR structural model where the conformational diversity is mainly an expression of lack of structural constraints.

Improvements are also observed for CamShift, with RMSD-decreases of 0.3–1.7 and 0.2 ppm for heavy and hydrogen atoms, respectively. In the case of PPM_One, Sparta+, and shAIC modest (up to 0.3 ppm) RMSD-decreases are observed for some ensembles but not others and, on average, the RMSD is roughly equally likely to remain unchanged or increase slightly. Finally, for ShiftX2 the RMSD consistently increases (by up to 0.7 ppm) on going from the X-ray structure to the ensembles, with the exception of C$\alpha$ where the RMSD is lowered by 0.1 ppm. We note that the RMSD values predicted with CamShift using the crystal structure are significantly larger than when using the CHARMM/CMAP structure (presumably due to hydrogen being optimized placed in accordance to the CHARMM22 topology file in the CamShift training set) and that the reduction in RMSD on going to ensembles is at most 0.3 ppm relatively to these values. So, it appears that the use of ensemble structures does not lead to a significant increase in accuracy compared to using a single structure for *any* of the empirical methods, in contrast to ProCS15 and CheShift-2.

The observations are consistent with earlier observations (*Parker, Houk & Jensen, 2006*; *Sumowski et al., 2014*; *Vila, Baldoni & Scheraga, 2009*; *Christensen et al., 2013*) that the empirical NMR prediction methods tend to be significantly less sensitive to changes in protein structure compared to DFT-based chemical shift predictors or chemical shifts computed using QM methods.

## SUMMARY AND OUTLOOK

In this paper we present ProCS15: a program that computes the isotropic chemical shielding values of backbone atoms and C$\beta$ given a protein structure in less than a second.

Table 4 **Comparison of chemical shifts predicted using various methods to experimental values measured for ubiquitin corrected for random coil effects.** The RMSD values are computed after linear regression. The predictions are done using a single X-ray structure (1UBQ) and five NMR-derived ensembles of varying size (indicated in parentheses for 1UBQ) without further refinement of the structure.

| | C$\alpha$ | C$\beta$ | C$'$ | H$\alpha$ | H$^N$ | N |
|---|---|---|---|---|---|---|
| | RMSD ($r$) | RMSD ($r$) | RMSD ($r$) | RMSD ($r$) | RMSD ($r$) | RMSD ($r$) |
| **ProCS15** | | | | | | |
| 1UBQ (1) | 1.7 (0.74) | 2.0 (0.50) | 1.7 (0.85) | 0.3 (0.80) | 0.6 (0.94) | 3.7 (0.80) |
| 1D3Z (10) | 1.3 (0.81) | 1.7 (0.62) | 1.7 (0.76) | 0.3 (0.81) | 0.5 (0.66) | 3.2 (0.83) |
| 2K39 (116) | 1.1 (0.84) | 1.7 (0.52) | 1.7 (0.69) | 0.3 (0.81) | 0.5 (0.61) | 3.6 (0.69) |
| 1XQQ (128) | 1.1 (0.84) | 1.8 (0.49) | 1.6 (0.74) | 0.3 (0.82) | 0.5 (0.61) | 3.7 (0.73) |
| 2LJ5 (301) | 1.1 (0.86) | 1.7 (0.55) | 1.6 (0.69) | 0.3 (0.82) | 0.6 (0.58) | 3.6 (0.74) |
| 2KOX (640) | 1.0 (0.89) | 1.7 (0.56) | 1.6 (0.71) | 0.2 (0.86) | 0.5 (0.65) | 3.5 (0.78) |
| **CheShift-2** | | | | | | |
| 1UBQ | 1.9 (0.58) | 1.9 (0.47) | | | | |
| 1D3Z | 1.3 (0.76) | 1.3 (0.70) | | | | |
| 2K39 | 1.3 (0.80) | 1.5 (0.62) | | | | |
| 1XQQ | 1.3 (0.81) | 1.6 (0.56) | | | | |
| 2LJ5 | 1.2 (0.82) | 1.4 (0.65) | | | | |
| 2KOX | 1.2 (0.83) | 1.4 (0.66) | | | | |
| **CamShift** | | | | | | |
| 1UBQ | 1.7 (0.75) | 1.9 (0.58) | 1.2 (0.74) | 0.3 (0.71) | 0.6 (0.52) | 4.5 (0.54) |
| 1D3Z | 1.0 (0.87) | 1.2 (0.75) | 0.9 (0.85) | 0.3 (0.80) | 0.5 (0.70) | 2.7 (0.72) |
| 2K39 | 1.1 (0.84) | 1.2 (0.80) | 1.0 (0.83) | 0.2 (0.87) | 0.4 (0.73) | 2.9 (0.65) |
| 1XQQ | 1.1 (0.84) | 1.2 (0.77) | 0.9 (0.85) | 0.2 (0.87) | 0.5 (0.68) | 2.9 (0.64) |
| 2LJ5 | 1.0 (0.86) | 1.4 (0.68) | 0.9 (0.85) | 0.2 (0.87) | 0.5 (0.71) | 3.1 (0.59) |
| 2KOX | 1.0 (0.88) | 1.1 (0.78) | 0.9 (0.85) | 0.2 (0.85) | 0.4 (0.73) | 2.8 (0.67) |
| **PPM_One** | | | | | | |
| 1UBQ | 0.7 (0.94) | 1.1 (0.84) | 0.9 (0.85) | 0.2 (0.87) | 0.6 (0.49) | 2.2 (0.81) |
| 1D3Z | 0.6 (0.96) | 0.9 (0.88) | 0.8 (0.89) | 0.2 (0.89) | 0.4 (0.78) | 1.8 (0.89) |
| 2K39 | 0.8 (0.95) | 1.0 (0.88) | 0.8 (0.89) | 0.2 (0.92) | 0.4 (0.78) | 2.2 (0.81) |
| 1XQQ | 0.8 (0.91) | 1.1 (0.84) | 0.8 (0.88) | 0.2 (0.92) | 0.4 (0.73) | 2.2 (0.82) |
| 2LJ5 | 0.6 (0.95) | 0.9 (0.88) | 0.8 (0.89) | 0.2 (0.93) | 0.4 (0.74) | 2.1 (0.84) |
| 2KOX | 0.6 (0.96) | 0.9 (0.89) | 0.8 (0.89) | 0.2 (0.93) | 0.4 (0.78) | 2.0 (0.85) |
| **Sparta+** | | | | | | |
| 1UBQ | 0.7 (0.94) | 1.0 (0.85) | 0.9 (0.86) | 0.2 (0.85) | 0.6 (0.48) | 2.0 (0.84) |
| 1D3Z | 0.6 (0.95) | 0.9 (0.87) | 1.0 (0.83) | 0.2 (0.86) | 0.4 (0.77) | 1.8 (0.88) |
| 2K39 | 0.7 (0.95) | 0.9 (0.88) | 1.0 (0.85) | 0.2 (0.89) | 0.4 (0.78) | 2.2 (0.83) |
| 1XQQ | 0.7 (0.93) | 1.0 (0.86) | 1.0 (0.84) | 0.2 (0.92) | 0.4 (0.72) | 2.2 (0.82) |
| 2LJ5 | 0.6 (0.96) | 0.9 (0.88) | 1.0 (0.84) | 0.2 (0.91) | 0.4 (0.76) | 2.1 (0.84) |
| 2KOX | 0.6 (0.96) | 0.9 (0.89) | 1.0 (0.84) | 0.2 (0.91) | 0.4 (0.77) | 2.0 (0.86) |
| **shAIC** | | | | | | |
| 1UBQ | 0.7 (0.93) | 1.1 (0.83) | 0.8 (0.89) | 0.3 (0.82) | 0.5 (0.69) | 2.0 (0.84) |
| 1D3Z | 0.6 (0.95) | 1.0 (0.85) | 0.7 (0.91) | 0.2 (0.85) | 0.4 (0.77) | 1.8 (0.87) |
| 2K39 | 0.7 (0.94) | 1.0 (0.84) | 0.7 (0.92) | 0.2 (0.85) | 0.4 (0.78) | 2.1 (0.83) |
Table 4 (*continued*)

| | Cα | Cβ | C′ | Hα | H$^N$ | N |
|---|---|---|---|---|---|---|
| | RMSD (*r*) | RMSD (*r*) | RMSD (*r*) | RMSD (*r*) | RMSD (*r*) | RMSD (*r*) |
| 1XQQ | 0.7 (0.94) | 1.1 (0.80) | 0.7 (0.91) | 0.2 (0.85) | 0.4 (0.72) | 2.2 (0.82) |
| 2LJ5 | 0.6 (0.95) | 1.0 (0.86) | 0.7 (0.91) | 0.2 (0.86) | 0.4 (0.75) | 2.1 (0.84) |
| 2KOX | 0.7 (0.94) | 1.0 (0.85) | 0.7 (0.91) | 0.2 (0.85) | 0.4 (0.74) | 2.0 (0.86) |
| **ShiftX2** | | | | | | |
| 1UBQ | 0.5 (0.97) | 0.4 (0.97) | 0.4 (0.97) | 0.1 (0.99) | 0.1 (0.98) | 1.3 (0.94) |
| 1D3Z | 0.4 (0.98) | 0.7 (0.94) | 0.6 (0.95) | 0.1 (0.96) | 0.2 (0.93) | 1.6 (0.91) |
| 2K39 | 0.4 (0.98) | 0.7 (0.93) | 0.7 (0.93) | 0.1 (0.98) | 0.2 (0.92) | 2.1 (0.85) |
| 1XQQ | 0.5 (0.97) | 0.8 (0.91) | 0.7 (0.93) | 0.1 (0.99) | 0.3 (0.90) | 2.0 (0.86) |
| 2LJ5 | 0.4 (0.98) | 0.6 (0.95) | 0.7 (0.94) | 0.1 (0.98) | 0.3 (0.92) | 1.9 (0.87) |
| 2KOX | 0.4 (0.98) | 0.6 (0.95) | 0.7 (0.93) | 0.1 (0.98) | 0.2 (0.92) | 1.8 (0.88) |

ProCS accounts for the effect of backbone and side-chain dihedral angles of a residue and the two neighboring residues, hydrogen bonding to the backbone amide group and Hα as well as ring-current effects (*Christensen, Sauer & Jensen, 2011*) on the hydrogen atoms and assumes that these effects are additive. The backbone, side-chain and hydrogen bonding terms are based on ∼2.35 million OPBE/6-31G(d,p)//PM6 calculations on tripeptides and small structural models of hydrogen-bonding.

ProCS15 reproduces the chemical shielding values computed using PCM/OPBE/6-31G(d,p)//PM6-D3H+for Ubiquitin and GB3 with RMSD values (after linear regression) of up to 2.5 ppm for carbon atoms, 0.8 ppm for hydrogen atoms, and 4.5 ppm for nitrogen. These deviations, which presumably result from the assumption of additivity and the simplified model systems, does not appear to preclude equal or better accuracy in comparison to experiment because the accuracies of the chemical shifts computed using ProCS15 (based on linear regression of the chemical shifts, cf. Eq. (1)) are very similar to the corresponding DFT calculations using single Ubiquitin and GB3 structures. The largest RMSD values observed for carbon, hydrogen, and nitrogen are, respectively, 2.2 (2.8) ppm, 0.7 (0.6) ppm, and 4.7 (4.6) ppm for ProCS15 (PCM/OPBE/6-31G(d,p)). These accuracies are very similar to DFT-based predictions made by other researchers (e.g., *Zhu, He & Zhang, 2012*; *Zhu, Zhang & He, 2013*; *Exner et al., 2012*) as well as CheShift-2 (*Martin et al., 2013*), which is another DFT-based chemical shift predictor for Cα and Cβ atoms. The RMSD values computed using ProCS15 for Ubiquitin can be reduced by as much as 0.7, 0.1, and 0.5 ppm for carbon, hydrogen, and nitrogen by using NMR-derived structural ensembles. Similar increase in accuracy is also observed for CheShift-2 (for Cα and Cβ) while for empirical chemical shift predictors the increase in accuracy is at most 0.3 ppm.

The latter observation is another indication that empirical chemical shift predictors are less sensitive to small structural changes, which may make them less suitable for chemical shift-guided refinement of protein structure compared to DFT-based predictors. Christensen and co-workers (*2013*) have already demonstrated that this is the case for amide hydrogen bonding geometries using a previous incarnation of ProCS limited to

amide proton chemical shift predictions and we are now planning similar refinement studies using all backbone atoms and C$\beta$ chemical shifts.

ProCS15 is freely available at github.com/jensengroup/procs15 and all structures and DFT calculations, including the full NMR shielding tensors, are available at erda.dk/public/archives/YXJjaGl2ZS1TYk40VXo=/published-archive.html.

## ACKNOWLEDGEMENT

We thank Osvaldo Martin, Jorge Vila, and Xiao He for helpful comments.

### Funding

This work was supported by the Lundbeck Foundation and the Danish e-Infrastructure Cooporation. The funders had no role in study design, data collection and analysis, decision to publish, or preparation of the manuscript.

### Grant Disclosures

The following grant information was disclosed by the authors:
Lundbeck Foundation and the Danish e-Infrastructure Cooporation.

### Competing Interests

The authors declare there are no competing interests.

### Author Contributions

- Anders S. Larsen performed the experiments, analyzed the data, contributed reagents/materials/analysis tools, reviewed drafts of the paper.
- Lars A. Bratholm analyzed the data, contributed reagents/materials/analysis tools, reviewed drafts of the paper.
- Anders S. Christensen conceived and designed the experiments, reviewed drafts of the paper.
- Maher Channir performed the experiments, reviewed drafts of the paper.
- Jan H. Jensen conceived and designed the experiments, analyzed the data, wrote the paper, prepared figures and/or tables, reviewed drafts of the paper.

### Data Availability

ProCS15 is freely available at github.com/jensengroup/procs15.

All structures and DFT calculations, including the full NMR shielding tensors, are available at erda.dk/public/archives/YXJjaGl2ZS1TYk40VXo=/published-archive.html.

### Supplemental Information

Supplemental information for this article can be found online at http://dx.doi.org/10.7717/peerj.1344#supplemental-information.

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
