# Peer review of "ProCS15: a DFT-based chemical shift predictor for backbone and Cβ atoms in proteins"

_PeerJ, doi:10.7717/peerj.1344_

## Round 0.1 · original submission · Minor Revisions

· Academic Editor

Minor Revisions

Both reviewers have some minor corrects to make and the second reviewer raises a point of skepticism about QM-based vs empirical estimators. A discussion addressing this would likely be of benefit to the field.

·

Basic reporting

No Comments

Experimental design

No comments

Validity of the findings

No comments

Additional comments

This manuscript is of great importance and I totally support its publication in PeerJ. The authors present an excellent and accurate chemical shift prediction program (ProCS15) based on millions of DFT calculations on simplified models. ProCS15 has extended the capability of previous ProCS program, which predicts the backbone amide proton chemical shift, to fast estimation of chemical shifts of backbone and C beta atoms in large proteins. The accuracies of chemical shifts on two proteins (namely, Ubiquitin and GB3) predicted by ProCS15 are very close to the results from fragment-based DFT calculations by Zhu et al., and Exner and co-workers. Nevertheless, the computational cost of ProCS15 is within a second. This program will be widely used in the NMR community. I only have a few minor points.

1) In the Introduction section, “RMSD observed for QM-based chemical shift predictions may, at least in part, be due to relatively small errors in the protein structures used for the predictions, and not a deficiency in the underlying method.” I agree with the first half of the statement, however, the limitation of current density functionals also contributes to the discrepancy between experiment and DFT calculations, especially for the 15N chemical shift prediction.

2) The first AF-QM/MM work is highly recommended to be cited in the paper,
He X., Wang B. and Merz K.M., Protein NMR Chemical Shift Calculations Based on the Automated Fragmentation QM/MM Approach. J. Phys. Chem. B 113, 10380 (2009)

·

Basic reporting

No comments.

Experimental design

No comments

Validity of the findings

No comments

Additional comments

This work is a direct extension of the author’s previous work on quantum based protein chemical shift calculation. The performance is comparable to other quantum based predictors but is worse than current empirical predictors. Because of this, I am still skeptical about all quantum-based predictors. Without solid cross-validation, it is very hard to argue that quantum predictors can capture subtle effect better than empirical predictors. It is true they respond more sensitively to minor structural change, but not necessary in a correct way. On the other hand, it is very useful for the whole community to have more selections that is different from previous ones. (Note that predictions from most empirical predictors are highly correlated, i.e., it won’t provide more information by switching from one to another empirical predictor.) In this context, this work should be published.

It is nice that the prediction performance can be improved a lot if applied to more realistic NMR-derived ensembles. This is expected because the experimental chemical shift of a given nucleus reflects the Boltzmann-weighted average of the 'instantaneous' chemical shifts of a large number of conformational substates that interconvert on the millisecond timescale or faster. This behavior has been discussed many times in the literature. All Ubiquitin NMR structures cited in this work are generated specifically to be a more realistic presentation of protein ensemble in solutions, except 1D3Z. 1D3Z is a traditional NMR structure model, where NMR conformer “bundle” should not be confused with a dynamic ensemble representation of the protein. In these types of NMR models, the spread of atomic positions merely provides information about the uncertainties of the atomic positions with respect to the average structure and has no direct physical meaning. The author may need to provide more comments on this in their last section titled “Comparison to experimental chemical shifts using NMR-derived ensembles”.

---

## Round 0.2 · accepted · Accept

· Academic Editor

Accept

Congratulations! Good work.